# Characterisation of molecular motions in cryo-EM single-particle data by multi-body refinement in RELION

Takanori Nakane[1], Dari Kimanius[2], Erik Lindahl[2,3], Sjors HW Scheres[1]*

[1]MRC Laboratory of Molecular Biology, Cambridge, United Kingdom; [2]Department of Biochemistry and Biophysics, Science for Life Laboratory, Stockholm University, Stockholm, Sweden; [3]Swedish e-Science Research Center, KTH Royal Institute of Technology, Stockholm, Sweden

**Abstract** Macromolecular complexes that exhibit continuous forms of structural flexibility pose a challenge for many existing tools in cryo-EM single-particle analysis. We describe a new tool, called multi-body refinement, which models flexible complexes as a user-defined number of rigid bodies that move independently from each other. Using separate focused refinements with iteratively improved partial signal subtraction, the new tool generates improved reconstructions for each of the defined bodies in a fully automated manner. Moreover, using principal component analysis on the relative orientations of the bodies over all particle images in the data set, we generate movies that describe the most important motions in the data. Our results on two test cases, a cytoplasmic ribosome from *Plasmodium falciparum*, and the spliceosomal B-complex from yeast, illustrate how multi-body refinement can be useful to gain unique insights into the structure and dynamics of large and flexible macromolecular complexes.

DOI: https://doi.org/10.7554/eLife.36861.001

## Introduction

In electron cryo-microscopy (cryo-EM) single-particle analysis, biological macromolecules are embedded in a thin layer of vitrified buffer and imaged in a transmission electron microscope. In principle, this represents a single-molecule imaging technique that provides unique information about the structure of individual macromolecular complexes. However, because the electron dose needs to be carefully limited to reduce radiation damage, cryo-EM images are typically extremely noisy and one needs to combine projections of many molecules supposedly in the same state to reliably recover high-resolution information. In recent years, with the development of direct-electron detectors and improved image processing procedures, this technique has allowed structure determination of many macromolecular complexes with enough detail to allow de novo atomic modelling (*Fernandez-Leiro and Scheres, 2016*).

Because macromolecular complexes often undergo conformational transitions as part of their functional cycles, many cryo-EM samples contain mixtures of different conformations. This type of structural heterogeneity may co-exist with incomplete complex formation or samples that have not been purified to homogeneity. In order to achieve high-resolution reconstructions, the presence of multiple different structures in the data needs to be dealt with during cryo-EM single-particle analysis. Many popular image classification approaches are based on competitive refinement of a user-defined number of references. These methods effectively divide the data into a discrete number of subsets or classes, each of which is assumed to be structurally homogeneous, for example see *Heymann et al. (2004)* and *Gao et al. (2004)* for early applications. Particularly useful are so-called unsupervised classification approaches, which do not require prior knowledge about the structural

**\*For correspondence:**
scheres@mrc-lmb.cam.ac.uk

heterogeneity in the data. Unsupervised classification of a discrete number of three-dimensional states became possible with the introduction of maximum likelihood classification methods (*Scheres et al., 2007*), which have since then been implemented in multiple image processing packages (*Sorzano et al., 2004*; *Scheres, 2012b*; *Lyumkis et al., 2013*; *Punjani et al., 2017*; *Grant et al., 2018*). Using many classes, competitive multi-reference refinement approaches have also been used to deduce energy landscapes from Boltzmann distributions of the relative number of particle images assigned to each of the class for flexible molecular complexes like the 26S proteasome (*Haselbach et al., 2017*) and the spliceosome (*Haselbach et al., 2018*).

However, discrete classification approaches are ultimately not well suited when macromolecular complexes exhibit continuous molecular motions. In principle, an infinite amount of classes would be needed to describe a continuum, and given a finite data size the number of particle images per class would approach zero. When a limited number of classes is used instead, each class will still contain residual structural heterogeneity. Several approaches have been proposed to deal with continuous heterogeneity in cryo-EM data. Moreover, classification based on multi-reference refinement aims to optimise the metric used in the target function (e.g. a marginal likelihood), which is not necessarily the optimal classification to understand the conformational transitions in the data. An early approach to describe continuous heterogeneity used normal-mode analysis to deduce macromolecular motions from low-resolution maps (*Tama et al., 2002*), and such predictions have also been used to guide alignment and discrete classification of cryo-EM images (*Jin et al., 2014*). Normal-mode reparametrisation has also been used to describe continuous deviations from helical symmetry in filamentous protein assemblies (*Rohou and Grigorieff, 2014*). An approach that in principle allows one to extract any three-dimensional state along a continuum is based on manifold embedding (*Dashti et al., 2014*). In this approach, each particle image represents a point in a multi-dimensional hyperspace, and a continuous manifold is deduced from the cloud of all points in the data set. Moving along this manifold then represents moving along the continuum of conformational changes. The manifold would ideally be calculated from the particle images alone, but currently available methods require prior alignment of the particle images against a single consensus reference. This may limit its effectiveness in cases where orientational and conformational assignments are intertwined.

Perhaps, a favourable case of continuous structural heterogeneity is when the molecular motion can be described by two or more rigid bodies that maintain their own internal structure but differ in their relative orientations. This model relies on the observation that tertiary protein structure often remains relatively constant upon domain movement. In case of such rigid-body motions, masked or focused refinements provide an efficient way to obtain high-resolution reconstructions. In this approach, at every iteration of the refinement process one masks away all density from the reference structure that does not correspond to a user-defined part of the complex. Thereby, the variability in the orientation of that part relative to the rest of the complex is ignored, and the part can be reconstructed to higher resolution. This procedure allowed atomic modelling in the presence of continuous variability in the relative orientations of ribosomal subunits for the yeast mitochondrial ribosome (*Amunts et al., 2014*) and the *Plasmodium falciparum* cytoplasmic ribosome (*Wong et al., 2014*). Later, this approach was improved by partial signal subtraction, where density corresponding to the rest of the complex is subtracted from the experimental particle images prior to performing focused refinements (*Bai et al., 2015*; *Zhou et al., 2015*; *Ilca et al., 2015*). Partial signal subtraction typically uses a so-called consensus refinement of all particle images against a single reference, and subtracts projections of the resulting consensus reconstruction in the directions of the consensus orientations from the experimental images. A limitation of the focused refinement approach is that each domain that is refined separately needs to be large enough to allow alignment of the individual subtracted particle images. To overcome this problem, a new approach called WarpCraft was described recently for the structure determination of transcription pre-initiation complexes with TFIIH and Mediator (*Schilbach et al., 2017*). WarpCraft uses normal mode analysis on a pseudo-atomic model of the cryo-EM map to restrain the motions between different regions in the map, thereby allowing reconstructions of much smaller regions than in focused refinement. However, there will exist a balance between separating highly flexible structures into many pieces and the amount of data available to reconstruct each piece.

In general, dealing with both discrete and continuous structural heterogeneity in cryo-EM data sets not only allows one to obtain higher-resolution maps, but also provides unique insights into the conformational landscape of macromolecular complexes. The presence of continuous forms of

structural heterogeneity represents one of the most important open questions in cryo-EM single-particle analysis. Nowadays, many groups apply combinations of different approaches to disentangle structural heterogeneity in high-resolution cryo-EM data sets, and optimal results depend strongly on user expertise in designing this strategy (*Fernandez-Leiro and Scheres, 2016*).

Here, we introduce a new approach to describe continuous structural heterogeneity in cryo-EM single-particle data in a user-friendly and fully automated manner. This approach, called multi-body refinement, builds on the approach of focused refinement with partial signal subtraction, and relies on the user to divide the reconstructed map from a consensus refinement into a discrete number of independently moving bodies. The multi-body model provides a balance between being able to describe large conformational transitions, while still being able to average (parts of) projections from the entire data set. During every iteration of multi-body refinement, the best relative orientation of each body is determined for every particle image, while the signal from the other bodies is subtracted on-the-fly. By keeping track of the relative orientations of all bodies for each particle image from the previous iteration, the partial signal subtraction is continually improved. Moreover, we use the refined relative orientations of all bodies upon convergence of the multi-body refinement to generate movies of the principal motions that exist within the data set. We first used an early implementation of our multi-body refinement to improve the density of mobile domains in the spliceosomal tri-snRNP complex (*Nguyen et al., 2015*). Since then, an approach that updates the relative orientations of different parts of a molecule was also implemented by others (*Schoebel et al., 2017*). In this paper, we formally introduce the multi-body approach and its implementation in RELION, and describe its application to two test cases: the cytoplasmic ribosome (*Wong et al., 2014*) from *Plasmodium falciparum*, and the spliceosomal B-complex from yeast (*Plaschka et al., 2017*). These results showcase how multi-body refinement can improve cryo-EM reconstructions and provide insights into the conformational landscape of large and flexible macromolecular complexes.

## Materials and methods

**Key resources table**

| Reagent type | Designation | Reference | Identifier |
|---|---|---|---|
| software | RELION | (*Scheres, 2012b*) | RRID:SCR_016274 |

### Theoretical background

Multi-body refinement in RELION is based on the assumption that all particles in a cryo-EM data set comprise the same macromolecular complex, that is, it is stoichiometrically homogeneous. Where multi-body refinement deviates from the standard approach for refinement of a single structure (or class) in RELION (*Scheres, 2012a*), is in the assumption that the macromolecular complex of interest behaves as $B$ separate rigid bodies. These bodies are identical in all particles, but their relative orientations are permitted to vary among the particles.

In Fourier space, the model is described as follows:

$$X_i = \mathrm{CTF}_i \left( \sum_{b=1}^{B} \mathbf{P}_{\phi_b} V_b \right) + N_i \,, \tag{1}$$

where:

- $X_i$ is the 2D Fourier transform of the projection image of the $i$th particle, with $i = 1, \ldots, N$.
- $\mathrm{CTF}_i$ is the 2D contrast transfer function for $X_i$.
- $V_b$ is the 3D Fourier transform of the $b$th rigid body. Its 3D Fourier components are assumed to be independent, zero-mean, Gaussian-distributed with variance $\tau_b^2$, which varies with spatial frequency.
- $\mathbf{P}_{\phi_b}$ represents the operation that extracts a slice out of the 3D Fourier transform of the $b$th body, and $\phi_b$ defines the orientation of that body with respect to the particle, comprising a 3D rotation and a phase shift according to a 2D translation in the image plane.
- $N_i$ is independent, zero-mean Gaussian noise in the 2D complex plane with variance $\sigma_i^2$, which varies with spatial frequency.

In analogy to standard refinement in RELION, optimisation of a regularised, marginal likelihood function is performed using expectation-maximisation (**Scheres, 2012b**). Because the sum of the $B$ bodies represents the same underlying 3D structure as in the standard refinement, the main difference in multi-body refinement is a $B$-fold increase in the number of hidden variables $\phi_b$ to describe the orientations of all bodies in every particle image. Based on the model in **Equation (1)**, the likelihood $P(X_i|\phi, \Theta^{(n)})$ of observing the $i$th experimental image given the current model parameter set $\Theta^{(n)}$ and any combination $\phi$ of all orientations $\phi_b$ can be calculated as a multivariate Gaussian centred on the difference between the particle image and the corresponding reference projection:

$$P\left(X_i|\phi,\Theta^{(n)}\right) \propto exp\left(\left|\frac{X_i - \mathrm{CTF}_i \sum_{b=1}^{B}\mathbf{P}_{\phi_b}V_b}{\sigma_i^2}\right|^2\right). \tag{2}$$

To prevent having to integrate for each body $b$ over the orientations $\phi_{b'}$ of all the other bodies $b' \neq b$, we treat each body $b$ separately and assume that the most likely orientations of the other bodies as determined for that particle image in the previous iteration ($\phi_{ib'}^{\star}$) are the correct ones. Thereby, we can rewrite **Eq (2)** as:

$$P\left(X_i|\phi_b,\Theta^{(n)}\right) \propto exp\left(\left|\frac{S_{ib} - \mathrm{CTF}_i\mathbf{P}_{\phi_b}V_b}{\sigma_i^2}\right|^2\right), \tag{3}$$

where $S_{ib}$ is the $i$th particle image, from which the CTF-modulated reference projections of all the other bodies $b' \neq b$ have been subtracted:

$$S_{ib} = X_i - \mathrm{CTF}_i\sum_{b'\neq b}^{B}\mathbf{P}_{\phi_{ib'}^{\star}}V_{b'} \tag{4}$$

By calculating all $S_{ib}$ on-the-fly during every expectation step, we can use **Eq (3)** to obtain posterior distributions $\Gamma_{i\phi_b}$ of all orientations $\phi_b$ being the correct one for the $i$th particle image given the model estimates at the current iteration $(n)$:

$$\Gamma_{i\phi_b}^{(n)} = \frac{P\left(\mathbf{X}_i|\phi_b,\Theta^{(n)}\right)P\left(\phi_b|\Theta^{(n)}\right)}{\int_{\phi_b'} P\left(\mathbf{X}_i|\phi_b',\Theta^{(n)}\right)P\left(\phi_b'|\Theta^{(n)}\right)d\phi_b'}, \tag{5}$$

where $P\left(\phi_b|\Theta^{(n)}\right)$ expresses prior information about $\phi_b$. In our current implementation, $P\left(\phi_b|\Theta^{(n)}\right)$ is implemented as a Gaussian function centred on the rotations and translations of the consensus refinement (see below), and with user-defined standard deviations.

During the maximisation step, the posterior distributions are used to obtain updated estimates for each body $V_b$, and the optimal orientations $\phi_{ib}^{\star}$ of all bodies in every particle image using:

$$V_b^{(n+1)} = \frac{\sum_{i=1}^{N}\int_{\phi_b}\Gamma_{i\phi_b}^{(n)}\mathbf{P}_{\phi_b}^{\mathrm{T}}\frac{\mathrm{CTF}_iS_{ib}}{\sigma_i^2}d\phi_b}{\sum_{i=1}^{N}\int_{\phi_b}\Gamma_{i\phi_b}^{(n)}\mathbf{P}_{\phi_b}^{\mathrm{T}}\frac{\mathrm{CTF}_i^2}{\sigma_i^2}d\phi_b + \frac{1}{\tau_b^2}}, \tag{6}$$

$$\phi_{ib}^{\star(n+1)} = \max_{\phi_b}\Gamma_{i\phi_b}^{(n)}. \tag{7}$$

Thereby, multi-body refinement is closely related to focused refinement with partial signal subtraction (**Bai et al., 2015**). However, whereas partial signal subtraction is typically performed once with a consensus model prior to starting a focused refinement, in multi-body refinement the partial signal subtraction is performed at every iteration with updated estimates for the reconstructed bodies and their relative orientations. A schematic overview of the multi-body approach is shown in **Figure 1**.

## Implementation details

Multi-body refinement has been implemented as a continuation of a standard 3D auto-refine job (in the program relion_refine_mpi) and is hardware-accelerated on both CPU or GPU (**Kimanius et al.,**

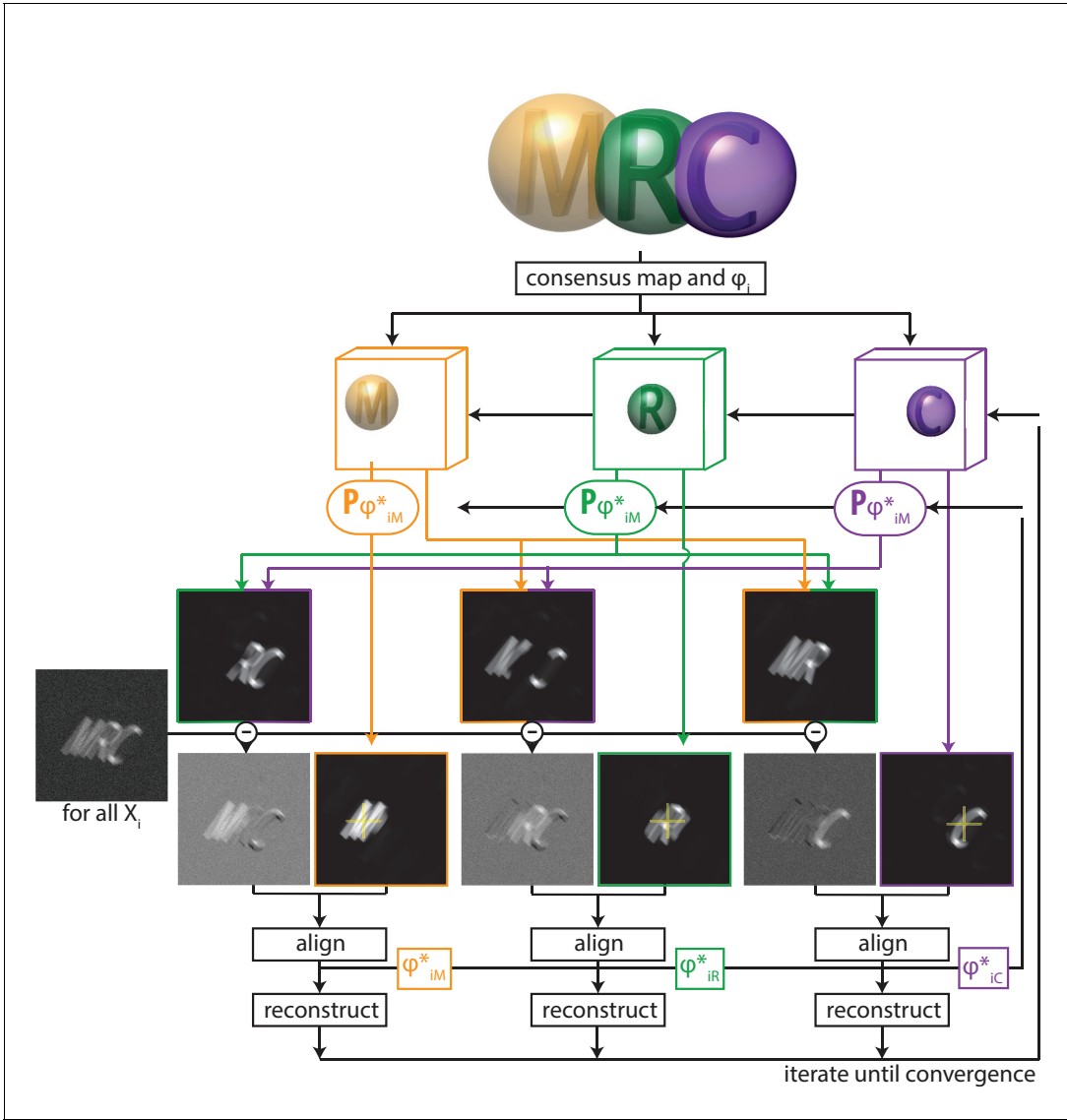

**Figure 1.** A schematic overview of multi-body refinement. After a consensus 3D auto-refinement in RELION, the three-dimensional consensus map is split into $B$ separate bodies using user-defined masks. In this example, $B = 3$ and the letters 'M' (orange), 'R' (green) and 'C' (puple) each represent a body, and the corresponding spherical masks are shown with transparency. During multi-body refinement, one performs focused refinement for all experimental particle images $X_i$, with local rotational and translational searches around the orientations from the consensus refinement. The yellow crosses in the reference projections for the focused refinements of each body indicate the centre-of-mass of each body's mask, around which all rotations are made.For each body, partial signal subtraction is performed with projections along the current estimates for the respective orientations of the other $B - 1$ bodies. This leads to $B$ subtracted versions of each experimental particle image during every iteration, which are aligned against projections of the corresponding body. The resulting optimal orientations $\phi_i^*$ for each body are used for the partial signal subtraction in the next iteration. Iterative alignment and reconstruction of all three bodiesis is repeated until convergence, which is when resolutions no longer improve and changes in the relative orientations of all bodies become small.

DOI: https://doi.org/10.7554/eLife.36861.002

The following source data and figure supplements are available for figure 1:

**Source data 1.** The body definition STAR file.
DOI: https://doi.org/10.7554/eLife.36861.005
**Figure supplement 1.** Overlapping body masks.
DOI: https://doi.org/10.7554/eLife.36861.003
**Figure supplement 2.** Relative body orientations.
DOI: https://doi.org/10.7554/eLife.36861.004

*2016*). We will refer to the previous standard refinement as the consensus refinement. It provides an initial estimate for the reconstructed density of each body, relative orientations for all particle images with respect to the consensus reconstruction, and estimates for the resolution-dependent variance in the experimental noise $\sigma_i^2$.

The definition of the $B$ separate bodies is provided by the user through a dedicated metadata file in STAR format (*Figure 1—source data 1*). The table in this file contains a single row for each body, where the entry rlnBodyMaskName points to a real-space mask that defines the outline of the corresponding body. Because sharp edges on masks cause artefacts in the Fourier-space refinements inside RELION, the user should provide masks with soft edges, i.e. they contain grey values between zero outside the body, and one inside the body. Although *Equation (1)* only holds for non-overlapping masks, our actual implementation also allows for overlapping masks between the different bodies, and the program will subtract only the non-overlapping parts of the masks of the other bodies when calculating $S_i^b$ for the focused refinements. For example, in *Figure 1*, when calculating the subtracted image for the first body, the program will only subtract projections of the second body corresponding to the volume that does not overlap with the first body; and of the volume of the third body that does not overlap with the first or the second body (*Figure 1—figure supplement 2A*). Likewise, when calculating subtracted images for the second body, only the volume of the first body that does not overlap with the second will be subtracted, and only the volume of the third body that does not overlap with the first or second body will be subtracted, etc. Because bodies that are higher up in the STAR file are subtracted first, and only non-overlapping parts of subsequent bodies are subtracted, one should position larger bodies above smaller ones in the STAR file. This treatment of overlapping bodies results in an overhead in required computer memory, as besides the 3D Fourier transforms of the $B$ bodies, one also needs to store Fourier transforms of their non-overlapping parts. Therefore, if all bodies overlap with all other bodies, $B^2$ 3D Fourier transforms are stored in memory (see *Figure 1—figure supplement 2B*).

The optimal orientations $\phi_{ib}^{*(n+1)}$ for all bodies of all particle images are stored in additional tables (called data_images_body_b, with $b$ being the body number) in the _data.star file that is written out at every iteration. These orientations are defined as residual rotations and translations that need to be applied on top of the orientations from the consensus refinement. Internally, every body is reconstructed with the centre-of-mass of its mask in the centre of the particle image box, and rotations are performed around this centre. The entry called rlnBodyRotateRelativeTo in the bodies STAR file defines the vector around which the rotations of the corresponding body are made: from its own centre-of-mass to the centre-of-mass of the body defined by this entry. For example, according to the STAR file defined in *Figure 1—source data 1*, the letters 'M' and 'R' rotate relative to each other, while the letter 'C' rotates relative to the letter 'R'. This description of rotations permits meaningful priors, $P(\phi_b|\Theta^{(n)})$, for the residual rotations of the bodies: the entries rlnBodySigmaAngles define the standard deviation (10 degrees for the bodies defined in *Figure 1—source data 1*) of a Gaussian-shaped prior on the Euler angles of the three bodies. Thereby, larger rotations are downweighted and rotational searches are limited from −30 to +30 degrees (i.e. ±3 times the standard deviation) around the consensus rotation for each of the three Euler angles. To avoid ambiguities in rotations where the second Euler angle (rlnAngleTilt) is close to zero, the stored Euler angles represent a rotation around a vector that is orthogonal to the vector between the centres-of-mass of the bodies. Thereby, the residual rotations are all around the Euler-angle values rlnAngleRot = 0, rlnAngleTilt = 90 and rlnAnglePsi = 0 (see *Figure 1—figure supplement 3*). The entries for rlnBodySigmaOffset in the bodies STAR file define the standard deviation (in pixels) of a Gaussian prior on the translational offsets for each of the bodies, which again are relative to the translations for the entire particles as defined in the consensus refinement.

Running as a continuation of a consensus 3D auto-refinement, the multi-body refinement will estimate the power of the signal $\tau_b^2$ for each body from the Fourier shell correlation (FSC) between two independently-refined half-sets of the data. Because the individual bodies occupy a relatively small volume in the particle image box, the option "-–solvent_correct_fsc" is activated by default. This option performs an internal correction to the FSC curves that accounts for the convolution effects of the mask, much like the post-processing job-type does in the standard RELION approach (*Chen et al., 2013*). The multi-body refinement approach is started from user-defined initial sampling rates for the rotations and translations, which are automatically increased during the refinement process. At

every iteration the algorithm assesses whether the resolution is still increasing and whether the orientational assignments are still changing. The orientations of an individual body will be kept fixed once the angular sampling becomes finer than the estimated accuracy of the rotations for that body, and the algorithm converges once the resolutions no longer increase, changes in the orientations become small, and the sampling rate is finer than the angular accuracies of all bodies.

In case of extensive structural heterogeneity, the initial extent of the body masks will be hard to assess in blurry parts of the consensus reconstruction. In such cases, an initial multi-body refinement may be run with a relatively large mask that comprises the entire blurry region of each body. The resulting maps after the first multi-body refinement may then allow the definition of tighter masks for one or more of the bodies. To allow a second multi-body refinement to proceed from the higher resolution reconstructions of the bodies in the first multi-body refinement, one can then provide an optional column in the body STAR file called rlnBodyReferenceName, which points towards the initial reference map for each of the bodies. An example of this is given below for the spliceosome test case.

## Analysis of multi-body orientations

Besides potentially improved densities for the individual bodies, multi-body refinement also outputs the optimal orientations $\phi_{ib}^{*}$ for all bodies and for all particle images in the data set. This information can be used to assess the molecular flexibility in the macromolecular complex. To this end, we have implemented a program called relion_flex_analyse. This program has two main applications. Firstly, it can write out subtracted images which may be useful in subsequent focused refinements or classifications outside the framework of multi-body refinement. To allow smaller image sizes and more meaningful priors in the subsequent refinements, the subtracted images are centered on the projected centre-of-mass of the remaining density after subtraction.

Secondly, the relion_flex_analyse program can also perform a principal component analysis on the relative orientations of the bodies of all particle images in the data set. For the principal component analysis, the three Euler angles describing the relative orientations of the bodies are taken into account, together with the two translations in the projection plane. In order to compare the two translational offsets in different projection directions, they are converted into three translations on the three-dimensional Cartesian grid of the reconstructions by setting the translations along the projection direction to zero. Thereby, principal component analysis is performed on six variables per body. The corresponding columns in the principal component analysis are normalised by the squared difference of the intensity values in the maps for each of the bodies after rotating them one degree or translating them one pixel in each of the directions. For each specified eigenvector, the program then outputs a user-specified number of maps ($M = 10$ by default). This is done by dividing the histogram of the amplitudes along the eigenvector of interest into $M$ equi-populated bins, that is each bin contains $1/M$th of the particle images. For each bin, the same reconstructed densities of the $B$ bodies are positioned relative to each other according to the rotations and translations that correspond to the centre amplitude of that bin, and a combined map is generated by adding all repositioned body densities together. This generates $M$ combined maps for the entire complex with different relative orientations of the bodies, each corresponding to the median orientations for $1/M$th of the particle images in the data set. These maps can then be used to generate a movie that visualises the motion along that eigenvector. Movies can be made, for example, using the 'Volume Series' utility in UCSF Chimera (*Pettersen et al., 2004*) or using CueMol (www.cuemol.org), which we used to generate the images and movies in this paper. When analysing these movies, it is useful to consider that the principal components describe the largest variations in the data along orthogonal degrees of freedom, and that general motions in the data are formed by linear combinations of multiple principal components. Alternatively, the program can also be used to output a subset of particle images within a user-specified range of amplitudes along one of the eigenvectors. The latter can be used to classify particle images based on the extent of a specific motion in the data set. An example of the latter is shown below for the ribosome test case.

## Results

### A ribosome test case

The multi-body refinement approach was tested on two previously published, experimental data sets. The first data set comprises 105,247 particle images of a *Plasmodium falciparum* ribosome bound to the drug emetine (*Wong et al., 2014*). This data set is available as entry 10028 on the EMPIAR data base (*Iudin et al., 2016*) and is used as a standard benchmark for RELION. We used the so-called polished particle images, that is particle images after movie-refinement and radiation damage weighting in RELION (*Scheres, 2014*).

The consensus refinement was started from a 60 Å low-pass filtered version of EMDB entry 2660, and yielded an overall resolution estimate of 3.2 Å after standard RELION post-processing to account for the solvent effects of the mask on the FSC curve (*Chen et al., 2013*). The estimated B-factor for sharpening this map was $-61$ Å$^2$ (*Rosenthal and Henderson, 2003*). In accordance with results described previously (*Wong et al., 2014*), the consensus map showed excellent density for the large ribosomal subunit, but the density was worse in the small subunit. In particular, the so-called head region of the small subunit exhibited much more fuzzy density than the rest of the complex. Therefore, we decided to split the ribosome into three bodies, which we named 'LSU' for the large subunit; 'SSU' for the small subunit without the head; and 'head' for the head region (*Figure 2A*). The corresponding masks were made using a 30 Å low-pass filtered version of the consensus map to define the boundary with the solvent region, and relied on available atomic models (PDB entries 3J79 and 3J7A) to determine the boundaries between the bodies. By placing soft-edges with a width of 11 Å on the boundaries of the masks, all three bodies overlapped with each other.

In a first multi-body refinement, the standard deviation of the Gaussian prior on the rotations was set to 10 degrees for all three bodies, and the standard deviations on the body translations were all set to two pixels (2.7 Å). These values represent an estimate of the amount of flexibility present in the different domains. Based on the domain architecture of the ribosome, the LSU and SSU were set to rotate with respect to each other, while the head was rotating with respect to the SSU. Multi-body refinement was started using an initial angular sampling rate of 1.8 degrees and an initial translational sampling rate of 0.25 pixels (0.33 Å). The choice for the initial angular sampling rate reflects a compromise between computational cost and a sufficiently fine sampling to describe the estimated flexibility. The initial translational sampling rate is similar to the estimated accuracy of the translations in the consensus refinement. Convergence of the multi-body refinement occured after 16 iterations, which took 16 hr on a single GPU work station with four NVIDIA 1080Ti GPUs, a 3.2 GHz Intel Xeon CPU, and 256 GB of RAM. Upon convergence, the solvent-corrected resolution estimates for the three bodies were: 3.1 Å for the LSU, 3.2 Å for the SSU, and 3.7 Å for the head. To test the influence of the mask boundaries on the results, we repeated the multi-body refinement with masks where the boundaries between the three bodies were defined by spheres that were manually positioned using the Volume Eraser tool in UCSF Chimera (*Pettersen et al., 2004*) to approximate the boundaries between the LSU, SSU and head. To test the influence of the standard deviations of the priors on the rotations and translations of the bodies, we also repeated multi-body refinement with standard deviations of 5 and 20 degrees on the rotations, and standard deviations of 1 pixel and 5 pixels on the translations of all three bodies. For all repeated multi-body refinements, the estimated resolutions in the three bodies did not differ by more than a single resolution shell (<0.1 Å) from the first multi-body-refinement, indicating that the approach is relatively robust to the choice of these parameters.

To assess the improvement in reconstructed density after multi-body refinement, we post-processed the maps after the consensus refinement using the three body masks and compared the resulting maps with the post-processed maps of the three bodies after multi-body refinement (*Figure 2B,C*). The same B-factor of $-61$ Å$^2$ was applied to all maps. The improvements for the LSU and SSU were modest, with average resolution in the LSU improving from 3.2 to 3.1 Å, and in the SSU from 3.3 to 3.2 Å, and visual inspection of the maps did not reveal major improvements (not shown). The improvements for the head were larger. In this region, the average resolution improved from 4.0 to 3.7 Å, and the reconstructed density improved considerably upon visual inspection. In particular, in the region furthest away from the centre of the ribosome, the reconstructed density for the head improved to such an extent that previously unmodelled regions became interpretable.

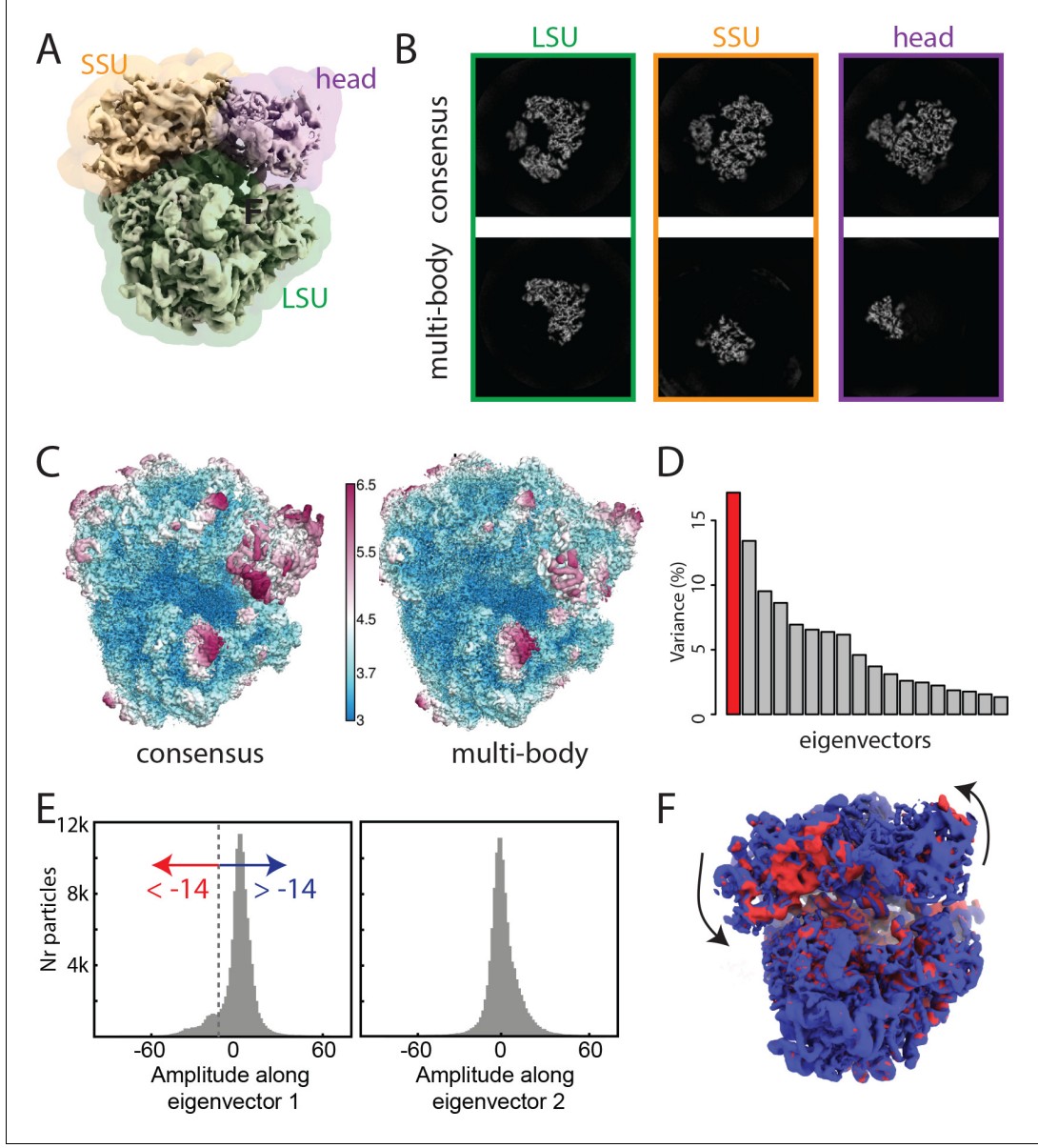

**Figure 2.** The ribosome test case. (**A**) The ribosome consensus map with the three transparent body masks (LSU, SSU and head) superimposed. (**B**) Slices through the density of the three bodies after the consensus refinement (top) and after multi-body refinement (down). (**C**) Local resolution estimates (in Å) calculated in RELION after the consensus refinement (left) and after multi-body refinement (right). (**D**) The contributions of all eigenvectors to the variance. The first eigenvector, for which the maps at the extremes are shown in panel F, is highlighted in red. (**E**) Histograms of the amplitudes along the first and second eigenvectors for all particle images in the data set. The histogram of the amplitudes along the first eigenvector shows a bimodal distribution. The data set was split into two subsets: particle images with the amplitude along the first eigenvector smaller than −14 (red arrow) and particle images with that amplitude larger than −14 (blue arrow). (**F**) Refined maps for the two subsets in the same colors. As observed in the movie along the first eigenvector, the SSU rolls with respect to the the LSU and the head swivels with respect to the SSU, *cf Video 1*.

DOI: https://doi.org/10.7554/eLife.36861.006

The following figure supplement is available for figure 2:

**Figure supplement 1.** Fourier shell correlation curves calculated from independently refined halves of the data for the three bodies after consensus refinement (dashed lines) and after the second multi-body refinement (solid lines).

DOI: https://doi.org/10.7554/eLife.36861.007

Application of the principal component analysis in the relion_flex_analyse program revealed that approximately 30% of the variance in the rotations and translations of the three bodies is explained by the first two eigenvectors (*Figure 2D*). Movies of the reconstructed body densities repositioned along these eigenvectors reveal that the first eigenvector corresponds to a rolling-like motion of the SSU with respect to the LSU and a concomittant swiveling of the head (*Video 1*), whereas the motion along the second eigenvector is more reminiscent of a ratchet-like motion of the SSU with respect to the LSU together with a displacement of the head (*Video 2*). With the exception of the amplitudes along the first eigenvector, histograms of all other amplitudes were monomodal. For the histogram of the amplitudes along the first eigenvector, a shoulder was visible on the negative side of the histogram, indicating that the structural heterogeneity along the first eigenvector may not be continuous. We then used the relion_flex_analyse program to write out two separate STAR files with 11,431 particle images for which the amplitude along the first eigenvector is less than $-14$, and 93,816 particle images for which the amplitude along the first eigenvector is greater than $-14$ (*Figure 2E*). Separate refinements of these subsets yielded overall resolution estimates of 4.4 and 3.2 Å, respectively. The differences between the two maps reveal similar differences in the orientation of the SSU and head as observed in *Video 1* (*Figure 2F*). This sort of discrete structural heterogeneity had remained undetected in our previous analysis (*Wong et al., 2014*), and re-running conventional 3D classifications on this data set (with six or eight classes, and with exhaustive, local or no alignments of the particle images after the consensus refinement; results not shown) did not yield class reconstructions with meaningful differences between them.

## A spliceosome test case

The second data set on which we tested multi-body refinement comprised 327,490 polished particle images of a spliceosomal B-complex from yeast (*Plaschka et al., 2017*). With the submission of this paper, we also submitted this data set to the EMPIAR data base, where it is now available under entry 10180. In the original study describing this data set (*Plaschka et al., 2017*), different parts of the complex were refined separately using focused refinement with partial image subtraction. Here, we used four masks that were generated in the original study for multi-body refinement: 'core' for the centre of the tri-snSNP structure, 'foot' for the tri-snSNP foot domains, 'helicase' for the helicase domain, and 'SF3b' for the SF3b subunits (*Figure 3A*). Note that the centre of the tri-snSNP structure is often called 'body' in the spliceosome literature, but we chose to call it the core to prevent confusion with the more general definition of body in this paper. The body masks were generated using a combination of the Volume Eraser tool in UCSF Chimera (*Pettersen et al., 2004*) and relion_mask_create. The masks were large enough to enclose the blurred, weak densities at the periphery of the complex, that is in the foot, helicase and SF3b domains. To reduce memory consumption, polished particle images used in the previous study (*Plaschka et al., 2017*) were down-sampled to 1.7 Å per pixel and re-windowed into a 320 pixel box. All refinements were performed without Fourier space padding by specifying the "—pad 1" option.

The consensus refinement was started from the B complex map in (*Plaschka et al., 2017*), which was low-pass filtered to 40 Å. The overall resolution after the consensus refinement, using a mask around the entire complex for post-processing, was 4.3 Å. The estimated B-factor for map sharpening was $-148$ Å$^2$. When postprocessed with individual masks for the four bodies, the core, foot, helicase and SF3b gave resolutions of 3.9, 4.2, 4.6 and 9.2 Å, respectively. Consistent with these values, the density for the SF3b was weak and blurred (*Figure 3C,D*).

For multi-body refinement, the core and the foot were set to rotate against each other. The helicase and the SF3b were set to rotate relative

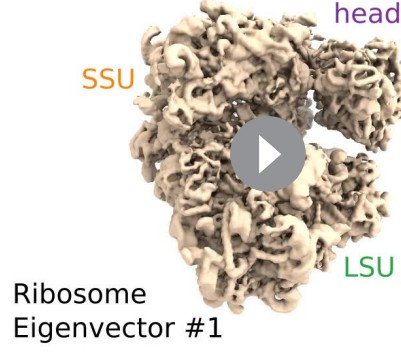

**Video 1.** Repositioning of the reconstructed body densities along the first eigenvector for the ribosome case reveals a rolling-like motion of the SSU with respect to the LSU and a concomitant swiveling of the head.

DOI: https://doi.org/10.7554/eLife.36861.008

to the core. Based on similar considerations as for the ribosome case, the initial angular sampling rate was set to 1.8 degrees, and the initial translational sampling range and rate were set to three pixel (5.1 Å) and 0.75 pixel (1.3 Å), respectively. The standard deviations of the angular and the translational prior were set to 10 degrees and two pixels (3.4 Å) for all bodies. The multi-body refinement converged in 15 iterations, which took 33 hr on the same GPU workstation as used for the ribosome case. Upon covergence, the estimated resolutions for the core, foot, helicase and SF3b domains were 3.8, 4.0, 4.5 and 5.1 Å, respectively.

Next, we ran a second round of multi-body refinement using tighter masks that were generated from the body reconstructions from the first multi-body refinement. These masks were generated by low-pass filtering the reconstruction for each body from the first multi-body refinement to 30 Å, extending the binarised maps by 10 pixels, and adding a soft edge of 5 pixels. In addition, we used 7 Å low-pass filtered maps of each body obtained in the first multi-body refinement as initial references for the second run using the rlnBodyReferenceName label in the body STAR file. The second multi-body refinement converged in 12 iterations, and took 40 hr on our GPU workstation. The resolutions of the core, foot and helicase remained essentially the same as in the first multi-body refinement (3.7, 4.0 and 4.5 Å, respectively), while that of the SF3b improved to 4.4 Å (*Figure 3C*; *Figure 3—figure supplement 1*). Despite the reasonable overall resolutions for each of the bodies, the densities for the U4 Sm ring within the helicase domain and the LSm ring and Rse1's $\beta$-propeller B (BPB) within the SF3b domain remained relatively fuzzy, indicating remaining flexibilities and/or compositional heterogeneity within these bodies.

In an attempt to address the remaining structural heterogeneity within the SF3b body, and as an illustration of how multi-body refinement can be combined with existing refinement approaches, we then used the relion_flex_analyse program to subtract the other three bodies from all experimental particle images, and performed a focused 3D classification on the SF3b without alignments. Using six classes, we identified a single class (containing 126,186 particle images) with better defined density than the other classes. A separate refinement of this class led to a SF3b map with improved local resolution of its central region (*Figure 3—figure supplement 2*) and an overall resolution of 4.0 Å. Subsequent focused refinements and classifications with partial signal subtraction on only the LSm ring did not yield better resolution maps (not shown).

We also examined the effect of wider search ranges for the rotations and translations of the bodies by changing the standard deviation of the angular and the translational priors to 20 degrees and 4 pixels. The resulting resolutions were within one or two resolution shells (<0.5Å) of those with the default values of 10 degrees and 2 pixels, which is consistent with the observations for the ribosome dataset.

Principal component analysis by relion_flex_analyse revealed that the first two components describe approximately 30% of the variance in the rotations and translations (*Figure 3D*). In contrast to the ribosome discussed above, the histograms were unimodal, suggesting that these motions were of a continuous nature. The first component corresponded to a rocking motion of the SF3b over the core (*Video 3*). The second represented concerted rocking of the helicase and SF3b (*Video 4*). Interestingly, the latter motion may resemble an early phase of the transition from the B complex (this structure) to the $B^{act}$ complex, where the SF3b regions moves towards the U6 snRNA ACAGAGA stem to form the spliceosome active site (*Yan et al., 2016*; *Rauhut et al., 2016*; *Plaschka et al., 2017*).

Finally, we compared the results from multi-body refinement with those from discrete classification. The two largest classes of a 3D classification run with eight classes (keeping the orientations fixed at those determined in the consensus refinement) comprised 40% and 27% of the particle images. These two classes represent a similar motion as identified by the first component in the relion_flex_analyse program (*Figure 3—figure supplement 3*). However, even for the largest class, the local resolutions for the SF3b, the core and the foot domain were lower than those obtained using multi-body refinement (*Figure 3—figure supplement 1*).

## Discussion

The main assumption in conventional cryo-EM single-particle analysis is that every experimental particle image is a two-dimensional projection of a common three-dimensional structure. This assumption no longer holds in the presence of continuous structural heterogeneity in the data set. Instead,

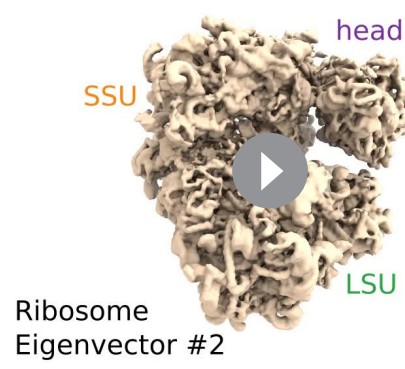

Ribosome
Eigenvector #2

**Video 2.** Repositioning of the reconstructed body densities along the second eigenvector for the ribosome case reveals a ratchet-like motion of the SSU with respect to the LSU together with a displacement of the head.
DOI: https://doi.org/10.7554/eLife.36861.009

multi-body refinement assumes that the continuous structural heterogeneity can be modelled as independent movements between rigid bodies, that is bodies that adopt the same structure but have different relative orientations among the data set. This assumption is probably reasonable for many macromolecular complexes, as tertiary protein structure often remains intact upon changes in quaternary structure. This is because within individual protein domains, the chemical environment of each amino acid does not change much upon movements of the entire domain. A major advantage of the multi-body model is that it allows one to use the entire data set for the reconstruction of each body, that is no subdivision of the data is required to describe the structural heterogeneity. The usefulness of this model is illustrated by the observation that multi-body refinement yields marked improvements over consensus refinement in the maps of flexible domains for both the ribosome and the spliceosome. For both test cases, parts of the consensus

map that were not interpretable in terms of an atomic model improved to resolutions around 4 Å after multi-body refinement. For the ribosome case, conventional 3D classification into a discrete number of classes turned out to be difficult and following standard procedures did not yield insights into the structural heterogeneity present. For the spliceosome case, multi-body refinement provided better maps than those obtained by conventional classification into a discrete number of classes. This is explained by the lower number of particle images that contribute to each class in discrete classification, whereas multi-body refinement allows the use of all particle images for each of the bodies. Moreover, in the presence of continuous heterogeneity, each discrete class will still correspond to a mixture of different structures. Therefore, we anticipate that multi-body refinement will be a useful tool in generating atomic models for domains that adopt multiple different orientations with respect to the rest of the complex.

Whereas the internal structure of protein domains may change little when a body moves relative to a neighbour, significant changes in the chemical environment are expected for amino acids at the interfaces between the bodies. As one domain moves relative to another, continuous or multiple discrete conformational changes may occur at the interface. Ultimately, these changes may also affect the internal structure of the protein domains, in which case multi-body refinement would no longer be justified, but in general the structural variability will be largest at the interfaces. Our implementation allows the definition of overlapping body masks. Thereby, each body can be defined to include the interface with its neighbouring bodies. Provided this interface density is not large enough to affect the particle image alignments, the resulting blurry density may help to better understand the nature of the structural variability at these interfaces.

Our implementation outputs a separate reconstruction for each of the bodies, which may then be used to build atomic models for the different bodies. Often, the experimentalist may want to combine these separately built atomic models into a single atomic model describing the entire complex. Such a combined atomic model can then be used to make figures for publication, or for submission to the Protein Data Bank. For example, in the original study on the spliceosomal B-complex, a combined atomic model was generated by rigid-body fitting the models that were built into maps resulting from separate focused refinements with partial signal subtraction, and a similar procedure could be used after multi-body refinement. However, we note that this representation of a single atomic model for the entire complex is in principle not supported by the data. Besides creating a false impression of structural homogeneity, in particular the conformations of residues at the interfaces of the rigid-body fitted atomic models may not reflect the true interface with the relative orientation of the bodies observed in the combined model. For example, a helix connecting the core of the spliceosome and the LSm ring of the SF3b body (marked by asterisks in *Figure 3D,E,G*) looks broken in

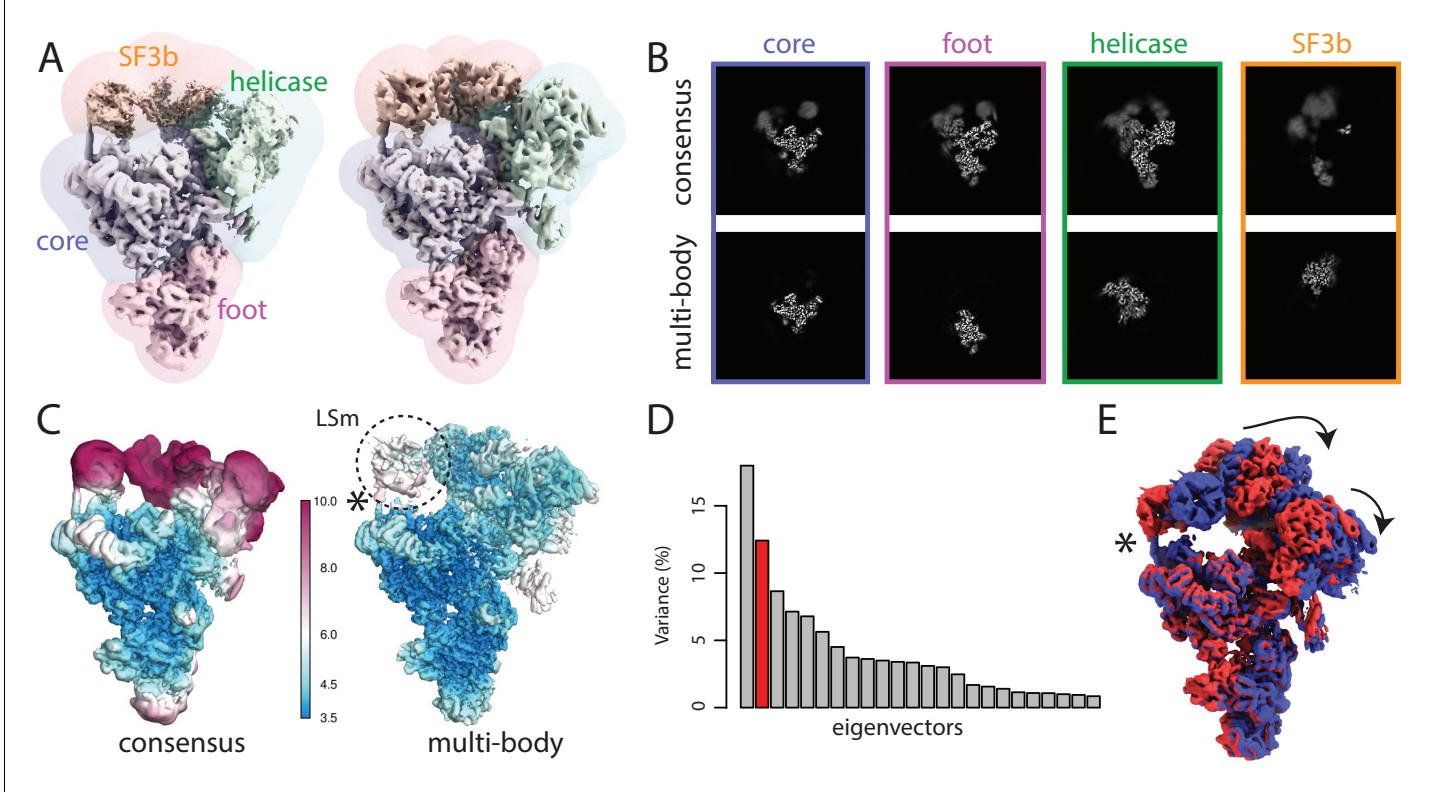

**Figure 3.** The spliceosome test case. (**A**) The four body masks used for the first spliceosome multi-body refinement are shown in semi-transparent colours on top of the consensus map on the left; the resulting density after the first multi-body refinement and the masks used for the second multi-body refinement are shown on the right. (**B**) Slices through the density of the four bodies after the consensus refinement (top) and after multi-body refinement (down). (**C**) Local resolution estimates (in Å) calculated in RELION after the consensus refinement (left) and after multi-body refinement (right). (**D**) The contributions of all eigenvectors to the variance. The second eigenvector, for which the maps at the extremes are shown in panel E, is highlighted in red. (**E**) Motion represented by the second eigenvector, *cf Video 4*. The helix that connects the core of the spliceosome and the LSm ring of the SF3b body that gets broken during the repositioning of the bodies in the eigenvector movies and the combined maps is indicated with an asterisk (also see main text).

DOI: https://doi.org/10.7554/eLife.36861.010

The following figure supplements are available for figure 3:

**Figure supplement 1.** Fourier shell correlation curves calculated from independently refined halves of the data for the four bodies after consensus refinement (dashed lines), for the largest class of a discrete 3D classification (dotted lines) and after the second multi-body refinement (solid lines).
DOI: https://doi.org/10.7554/eLife.36861.011

**Figure supplement 2.** Local resolution estimates (in Å) for the SF3b region after multi-body refinement (left) and after subsequent partial signal subtraction in relion_flex_analyse followed by focused classification and refinement of the best class (right).
DOI: https://doi.org/10.7554/eLife.36861.012

**Figure supplement 3.** The largest two classes of a 3D classification with eight classes (right) represent a similar motion as identified by the first principal component from the multi-body approach (left).
DOI: https://doi.org/10.7554/eLife.36861.013

combined multi-body reconstructions, which is chemically unfeasible. This is where the rigid body assumption no longer holds. In reality, the interface should rearrange to keep the helix intact. As many different interface structures may exist in the data set, reconstruction of all these different densities and the generation of atomic models for these may not be possible. One could propose to keep the atomic models for each of the bodies separate, as this would prevent the impression of a well-defined structure at the interface. However, such a solution would still not reflect the variability in conformations of residues at the interface, and would be more difficult to analyse by non-experts. How to tackle the problem of reflecting continuous structural heterogeneity with atomic models will require community-wide discussion.

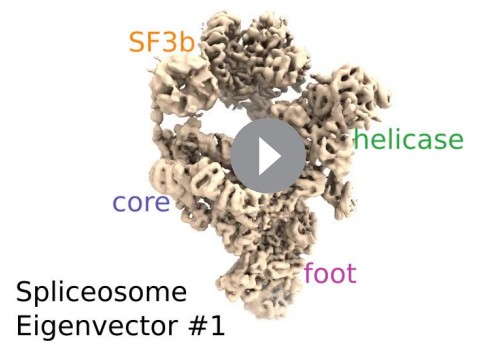

**Video 3.** Repositioning of the reconstructed body densities along the first eigenvector for the spliceosome case reveals a rocking motion of the SF3b domain over the core.
DOI: https://doi.org/10.7554/eLife.36861.014

Multi-body refinement improved the maps for flexible parts of both the ribosome and spliceosome test cases. Similar improvements can also be achieved by existing focused refinement or classification approaches with partial signal subtraction. In fact, in the original study describing the spliceosomal B-complex, parts of the SF3b region were reconstructed to higher resolutions (3.9 Å) than achieved in multi-body refinement (4.4 Å). Multi-body refinement has the advantage of iteratively improving the partial signal subtraction, whereas focused refinement and classification is typically done with fixed subtracted particle images. On the other hand, combining multiple focused classifications and/or refinement runs, each with its own user-specified sampling and mask parameters, is more flexible than the fully automated multi-body refinement, but requires more user expertise. To combine the advantages of both approaches, the relion_flex_-analyse program can output subtracted images according to the iteratively refined relative orientations from a multi-body refinement. The resulting subtracted particle images can then be used in subsequent focused classifications and/or refinements. An additional advantage of this program over previously existing subtraction tools in RELION is the centering at the projected centre-of-mass of the subtracted particle images, which allows the use of more meaningful priors and smaller box sizes. The use of this procedure was illustrated for the SF3b region of the spliceosome, where focused classification and refinement after subtraction in the relion_flex_analyse program led to a resolution of 4.0 Å, which is close to the resolution of 3.9 Å obtained for this region in the original study.

Besides improved reconstructed density for flexible domains, multi-body refinement also provides information about the relative orientations of all bodies for every particle image in the data set. The proposed principal component analysis on the relative rotations and translations of all bodies allows convenient visualisation of the principal motions that exist within the data set through the generation of movies. These movies provide the user with unique insights into how different bodies move with respect to each other, which bodies move together with others, etc. However, we do again point out that (as outlined above for the analysis of atomic models) densities at the interfaces between different bodies are not well-defined in these movies. For the ribosome, analysis of the movies for the first two eigenvectors revealed the presence of the typical ribosome motions of rolling and ratcheting of the SSU relative to the LSU. In addition, the presence of a bimodal histogram for the amplitudes along the first eigenvector indicated the presence of non-continuous heterogeneity in the rolling motion. In the general case of discrete heterogeneity, the movies generated from such non-unimodal histograms will display a discontinuous jump from one conformation to the other. Subsequent classification of the particle images based on the amplitude along the first eigenvector was also used to obtain two separately refined maps of the two states, which confirmed the motion observed along that eigenvector. For the spliceosome, rotations of the helicase and SF3b with respect to the core were observed. Interestingly, the

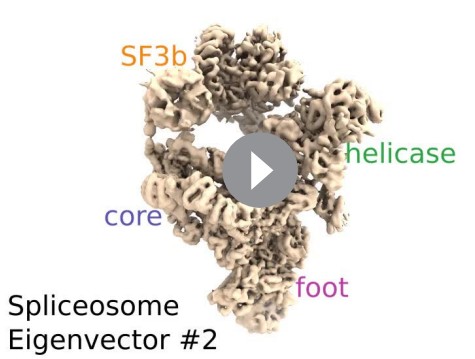

**Video 4.** Repositioning of the reconstructed body densities along the second eigenvector for the spliceosome case reveals a concerted rocking motion of the helicase and SF3b over the core.
DOI: https://doi.org/10.7554/eLife.36861.015

motion corresponding to the second eigenvector resembles an early phase of the transition from the B complex to the $B^{act}$ state in the spliceosome functional cycle (*Yan et al., 2016*; *Rauhut et al., 2016*; *Plaschka et al., 2017*). These results illustrate that the movies generated by the relion_flex_analyser program may be useful in exploring biologically relevant molecular motions.

One of the main limitations of multi-body refinement, and focused refinement in general, is that when the size of individual bodies decreases the signal becomes too weak for reliable alignment of the subtracted particle images. Refinement of macromolecular complexes with molecular weights below 100 kDa becomes progressively difficult, although smaller complexes have been refined, and phase-plate imaging (*Khoshouei et al., 2017*) can help to reduce the size limits. In the case of perfect partial signal subtraction, similar limits on the minimum size of the bodies are expected for multi-body refinement. In practice, partial signal subtraction will contain errors, and larger bodies may be required. The smallest body presented in this paper, the foot of the spliceosome, has a molecular weight of approximately 290 kDa. The current implementation allows keeping very small bodies fixed (by setting the standard deviations on their rotational and translational priors to zero). Thereby, at least the consensus density can be subtracted for the alignment of the other bodies.

Ultimately, one would aim to introduce dependencies between the alignments of bodies, such that simultaneous alignment of many small bodies would become feasible. This would allow modelling of much more complicated forms of continuous structural heterogeneity. The approach in Warp-Craft represents a step in that direction. WarpCraft could be considered as a many-body approach, where the flexibility is re-parametrised using normal mode analysis. This reduces the number of degrees of freedom from five per body (three Euler angles and two in-plane shifts) to the number of normal modes considered, regardless of the number of bodies. Consequently, WarpCraft may perform well in cases where the structural flexibility can be described by a few correlated local motions, as was probably the case for the Mediator complex (*Schilbach et al., 2017*). However, when multiple independently moving bodies are present, one again needs many normal modes to describe the entire conformational landscape, and the advantage of the normal-mode re-parametrisation would disappear. Moreover, the presence of domain rotations, which are highly non-linear deformations, as well as the presence of discrete heterogeneity are difficult to describe by normal modes. In such cases, provided the individual bodies are large enough, one would expect the multi-body approach to perform better than the approach implemented in WarpCraft. Probably, both approaches will suffer from similar problems at the boundaries between bodies that undergo large motions, as many data sets will not contain sufficient information to describe those. Currently, the exploration of new re-parametrisation schemes that allow modelling of commonly occurring types of flexibility in macromolecular complexes is a topic of active research in our groups.

Meanwhile, the multi-body approach presented here offers a convenient tool to improve the reconstructed density of flexible regions in macromolecular complexes that can be described as multiple moving rigid bodies, and to provide unique insights into the nature of these movements. The computer programs described in this paper will be distributed as part of release 3.0 of RELION, which is completely free for any user. As this software is distributed as open-source, others can contribute their own modifications and improvements of the presented algorithms, as has happened for partial signal subtraction approaches in the past (*Zhou et al., 2015*; *Ilca et al., 2015*; *Schoebel et al., 2017*). Based on the observations described here, we anticipate that multi-body refinement, possibly combined with existing classification and refinement approaches, will be a useful tool to extract more information from cryo-EM data sets on macromolecular complexes exhibiting continuous structural heterogeneity.

## Acknowledgements

We are grateful to Jake Grimmett and Toby Darling for assistance with computing. The ribosome data set in this paper was collected by Xiao-chen Bai from samples made by Wilson Wong. The spliceosome sample was made and the spliceosome data were collected by Clemens Plaschka and Pei-Chun Lin from the group of Kiyoshi Nagai. We thank Clemens Plaschka, Kiyoshi Nagai and Vish Chandrasekaran for helpful discussions and critical reading of the manuscript. This work was funded by the UK Medical Research Council through grant MC_UP_A025_1013 to SHWS and the Swedish Research Council through grant 2017–04641 to EL.

## Additional information

### Competing interests
Sjors HW Scheres: Reviewing editor, *eLife*. The other authors declare that no competing interests exist.

### Funding

| Funder | Grant reference number | Author |
| --- | --- | --- |
| Medical Research Council | MC UP A025 1013 | Sjors HW Scheres |
| Svenska Forskningsrådet Formas | 2017-04641 | Erik Lindahl |

The funders had no role in study design, data collection and interpretation, or the decision to submit the work for publication.

### Author contributions
Takanori Nakane, Software, Formal analysis, Validation, Investigation, Visualization, Methodology, Writing—original draft, Writing—review and editing; Dari Kimanius, Software, Investigation, Visualization, Methodology, Writing—review and editing; Erik Lindahl, Software, Supervision, Writing—review and editing; Sjors HW Scheres, Conceptualization, Software, Formal analysis, Supervision, Validation, Investigation, Visualization, Methodology, Writing—original draft, Project administration, Writing—review and editing

### Author ORCIDs
Takanori Nakane (iD) http://orcid.org/0000-0003-2697-2767
Dari Kimanius (iD) http://orcid.org/0000-0002-2662-6373
Erik Lindahl (iD) https://orcid.org/0000-0002-2734-2794
Sjors HW Scheres (iD) http://orcid.org/0000-0002-0462-6540

### Decision letter and Author response
Decision letter https://doi.org/10.7554/eLife.36861.021
Author response https://doi.org/10.7554/eLife.36861.022

## Additional files

### Data availability
The ribosome dataset was downloaded from the EMPIAR database (entry number 10028). The spliceosome dataset has been uploaded to the EMPIAR database as well, and is now available under entry number 10180.

The following dataset was generated:

| Author(s) | Year | Dataset title | Dataset URL | Database, license, and accessibility information |
| --- | --- | --- | --- | --- |
| Sjors HW Scheres | 2018 | Structure of a pre-catalytic spliceosome | https://www.ebi.ac.uk/pdbe/emdb/empiar/entry/10180/ | Publicly available at the Electron Microscopy Public Image Archive (accession no. EMPIAR-10180) |

The following previously published dataset was used:

| | | | | Database, license, and accessibility |
| --- | --- | --- | --- | --- |

| Author(s) | Year | Dataset title | Dataset URL | information |
|-----------|------|---------------|-------------|-------------|
| Sjors HW Scheres | 2015 | Cryo-EM structure of the Plasmodium falciparum 80S ribosome bound to the anti-protozoan drug emetine | http://dx.doi.org/10.6019/EMPIAR-10028 | Publicly available at the Electron Microscopy Public Image Archive (accession no. EMPIAR-10028) |

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
