## [Decision Letter]

Thank you for submitting your article "Characterisation of molecular motions in cryo-EM single-particle data by multi-body refinement in RELION" for consideration by *eLife*. Your article has been reviewed by three peer reviewers, and the evaluation has been overseen by Axel Brunger as the Reviewing Editor and Andrea Musacchio as the Senior Editor. The following individuals involved in review of your submission have agreed to reveal their identity: John L Rubinstein (Reviewer #2); Nikolaus Grigorieff (Reviewer #3).

The reviewers have discussed the reviews with one another and the Reviewing Editor has drafted this decision to help you prepare a revised submission.

Summary:

Scheres and colleagues describe an extension – multi-body refinement – of their software, RELION, to process images of single particles displaying large continuous heterogeneity that can be represented as motion of several independent rigid subdomains (the "bodies"). Their implementation follows a strategy that was applied previously, for example, to the ribosome and the spliceosome. The new tool is user-friendly and automates the steps established in these previous projects: masking of rigid subdomains by pre-defined masks, signal subtraction of parts of the particle outside the mask, focused refinement of the remaining domain, and combination of the separately refined domains to reconstitute the complete particle. In addition, the angles and translations for the subdomains can be analyzed using principal component analysis, and reconstructions with domains in intermediate positions can be calculated to visualize the motions. The new tool is applied to two previously published datasets, of Plasmodium ribosome and yeast spliceosome. Focused refinement itself is not new, but the new automated implementation in Relion should make it much easier to use when multiple focused regions (or multiple bodies) are necessary.

The multi-body refinement addresses an important problem in single-particle cryo-EM that occurs when domains of a complex move against each other without assuming distinct positions (conformations) relative to each other. This is one type of continuous heterogeneity that occurs frequently in large macromolecular machines. The authors demonstrate that multi-body refinement is an effective tool to improve the resolution of the subdomains. As they point out, features at the interfaces between the domains will not be reconstructed correctly.

Essential revisions:

The WarpCraft approach appears to be superior in preserving details at domain interfaces. The authors should discuss situations in which their multi-body approach would be preferred.

How does the visualization of the motion by principal component analysis work when the motion is not continuous, i.e. there are only a few discrete states? What if dividing the particles into *M* equi-populated bins according to the chosen eigenvalue is not possible because there are not sufficient particles for some bins?

The multi-body refinement results are compared with the consensus refinement results that were obtained by refining a single class. A more interesting comparison would be with the results obtained using traditional 3D classification assuming discrete states. In the ribosome example, this may yield results similar to the multi-body approach.

Some explanation of the rationale for choice of parameters subsection “A ribosome test case”and subsection “A spliceosome test case” would be valuable.

[Editors' note: further revisions were requested prior to acceptance, as described below.]

Thank you for resubmitting your work entitled "Characterisation of molecular motions in cryo-EM single-particle data by multi-body refinement in RELION" for further consideration at *eLife*. Your revised article has been favorably evaluated by Andrea Musacchio (Senior Editor), a Reviewing Editor, and three reviewers.

Reviewer #1:

I am satisfied with how the authors have addressed the concerns raised by the first round of reviews.

I caught a few typos.

Reviewer #2:

I am satisfied by the changes and think the paper should be published

Reviewer #3:

I read the authors' rebuttal and am happy with their replies. I think that the manuscript is ready for publication in *eLife*.

---

## [Author Response]

Essential revisions:The WarpCraft approach appears to be superior in preserving details at domain interfaces. The authors should discuss situations in which their multi-body approach would be preferred.

We thank the reviewers for pointing out that our manuscript apparently gave the false impression that the WarpCraft approach would be superior. We do not think this is the case. The WarpCraft approach is probably best suited for a different type of heterogeneity compared to the multi-body refinement in RELION, while none of the approaches is well suited to deal with heterogeneity at domain interfaces. To better describe the differences between the two methods, we have added the following paragraph to the Discussion section: "Ultimately, one would aim to introduce dependencies between the alignments of bodies, such that simultaneous alignment of many small bodies would become feasible. This would allow modelling of much more complicated forms of continuous structural heterogeneity. The approach in WarpCraft represents a step in that direction. WarpCraft could be considered as a many-body approach, where the flexibility is re-parametrised using normal mode analysis. This reduces the number of degrees of freedom from five per body (three Euler angles and two in-plane shifts) to the number of normal modes considered, regardless of the number of bodies. Consequently, WarpCraft may perform well in cases where the structural flexibility can be described by a few correlated, local motions, as was probably the case for the Mediator complex (Schilbach et al., 2017). However, when multiple independently moving bodies are present, one again needs many normal modes to describe the entire conformational landscape, and the advantage of the normal-mode re-parametrisation would disappear. Moreover, the presence of domain rotations, which are highly non-linear deformations, as well as the presence of discrete heterogeneity are difficult to describe by normal modes. In such cases, provided the individual bodies are large enough, one would expect the multi-body approach to perform better than the approach implemented in WarpCraft. Probably, both approaches will suffer from similar problems at the boundaries between bodies that undergo large motions, as many data sets will not contain sufficient information to describe those. Currently, the exploration of new re-parametrisation schemes that allow modelling of commonly occurring types of flexibility in macromolecular complexes is a topic of active research in our groups."

How does the visualization of the motion by principal component analysis work when the motion is not continuous, i.e. there are only a few discrete states?

In that case, the histogram of the amplitudes along the eigenvector becomes multi-modal, and the movie will contain a discontinuous jump from one confirmation to the other. To a limited extent, the ribosome case represents an example of this type of heterogeneity. In the Discussion section, we have modified the text below to reflect these observations.

"For the ribosome, analysis of the movies for the first two eigenvectors revealed the presence of the typical ribosome motions of rolling and ratcheting of the SSU relative to the LSU. In addition, the presence of a bimodal histogram for the amplitudes along the first eigenvector indicated the presence of non-continuous heterogeneity in the rolling motion. In the general case of discrete heterogeneity, the movies generated from such non-unimodal histograms will display a discontinuous jump from one conformation to the other."

What if dividing the particles into M equi-populated bins according to the chosen eigenvalue is not possible because there are not sufficient particles for some bins?

This question reveals a misunderstanding by the reviewer. As the bins are equi-populated, there are equal numbers of particles represented by each bin. E.g. for the default ten bins, there are always one-tenth of the data set in each bin. (Note that we have also modified the term 'eigenvalues' to the more correct 'amplitudes along the eigenvector'.) To prevent further misunderstandings by the readers, we have modified the following sentence in subsection “Analysis of multi-body orientations”:

"This is done by dividing the histogram of the amplitudes along the eigenvector of interest into *M* equi-populated bins, i.e. each bin contains 1/*M*th of the particle images. For each bin, the same reconstructed densities of the *B* bodies are positioned relative to each other according to the rotations and translations that correspond to the centre amplitude of that bin, and a combined map is generated by adding all repositioned body densities together."

The multi-body refinement results are compared with the consensus refinement results that were obtained by refining a single class. A more interesting comparison would be with the results obtained using traditional 3D classification assuming discrete states. In the ribosome example, this may yield results similar to the multi-body approach.

We have now performed standard discrete classifications for both the spliceosome and the ribosome. For the ribosome, 3D classification (using several runs) did not reveal any meaningful insights into the structural heterogeneity, which is in line with our observation when we originally published this data set. For the spliceosome, a discrete classification captured the overall motion observed in the first eigenvector of the multi-body analysis, but the resolution of the different bodies obtained in multi-body refinement is higher. In addition, the discrete classification did not yield insight in as many types of motions as detected by the principal components analysis of the multi-body approach. These new results have been added to Figure 3—figure supplement 1 and Figure 3—figure supplement 3. Apart from describing these observations in the Results section, we also added the following remark to the Discussion section:

"For the ribosome case, conventional 3D classification into a discrete number of classes turned out to be difficult and following standard procedures did not yield insights into the structural heterogeneity present. For the spliceosome case, multi-body refinement provided better maps than those obtained by conventional classification into a discrete number of classes. This is explained by the lower number of particle images that contribute to each class in discrete classification, whereas multi-body refinement allows the use of all particle images for each of the bodies. Moreover, in the presence of continuous heterogeneity. each discrete class will still correspond to a mixture of different structures."

Some explanation of the rationale for choice of parameters for subsection “A ribosome test case” and subsection “A spliceosome test case” would be valuable.

In subsection *“A ribosome test case”* we modified:

"In a first multi-body refinement, the standard deviation of the Gaussian prior on the rotations was set to 10 degrees for all three bodies, and the standard deviations on the body translations were all set to 2 pixels. These values represent an estimate of the total flexibility present in the different domains. Based on the domain architecture of the ribosome, the LSU and SSU were set to rotate with respect to each other, while the head was rotating with respect to the SSU. Multi-body refinement was started using an initial angular sampling rate of 1.8 degrees and an initial translational sampling rate of 0.25 pixels. The choice for the initial angular sampling rate reflects a compromise between computational cost and a sufficiently fine sampling to describe the estimated flexibility. The initial translational sampling rate is similar to the estimated accuracy of the translations in the consensus refinement."

In subsection “A spliceosome test case” we modified:

"Based on similar considerations as for the ribosome case, the initial angular sampling rate was set to 1.8 degrees, and the initial translational sampling range and rate were set to 3 and 0.75 pixels, respectively. The standard deviations of the angular and the translational prior were set to 10 degrees and 2 pixels for all bodies."